# An investigation of the impact of human capital and supply chain competitive drivers on firm performance in a developing country

**Ricardo Santa**[1]*, **Mario Ferrer**[2], **Thomas Tegethoff**[1], **Annibal Scavarda**[3]

**1** CESA–Colegio de Estudios Superiores de Administración, Santa Fé, Colombia, **2** Alfaisal University, Riyadh, Kingdom of Saudi Arabia, **3** Federal University of the State of Rio de Janeiro, Rio de Janeiro, Brazil

* ricardo.santa@cesa.edu.co

**Data Availability Statement:** Original Data used for the Structural Equation Modeling analysis have been uploaded to the Dryad repository in SPSS format. DOI: doi.org/10.7910/DVN/YGPBEV.

## Abstract

### Purpose

This paper aims to determine the effect that human capital and key competitive drivers such as quality, agility, and cost have on firm performance, whether this effect is related to the firm's outsourcing strategy, and whether the firm size is relevant in explaining such relationships.

### Design

This study uses structural equation modeling to test the hypothesized relationships for small to medium enterprises (SMEs) and large organizations using a custom survey based on a review of the literature and completed by 404 firms in the Valle del Cauca agro-industrial region in Colombia.

### Findings

Human capital strategies are essential for the effective deployment of operational agility, quality, and cost management strategies, which impact firm performance through effective outsourcing strategies. These relationships, however, do not hold the same across firms of different sizes. Specifically, outsourcing practices are lacking amongst SMEs in the studied region. The study is limited to a specific region, with infrastructure and connectivity limitations that hinder or undermine otherwise potentially valuable third-party logistics strategies.

### Practical implications

This paper contributes to the theory and practice in supply chain competitiveness by extending current knowledge of the impact of human capital and key competitive drivers on firm performance, highlighting regional specificities that could hinder firms' competitiveness, and by presenting a novel, quantitative methodology seldom used for these topics.

**Funding:** The authors received no specific funding for this study.

**Competing interests:** The authors have declared that no competing interests exist.

## 1. Introduction

To be competitive, firms implement value-adding strategies that build on key competitive drivers [1–3]. It follows that to better understand competitiveness at the firm level, we should identify these key drivers and assess their impact on performance. We are particularly interested in assessing how strategically deploying a skilled workforce (known as human capital) influences key competitive drivers such as an outsourcing strategy, operational agility, quality, and costs. This study tests the effect of these drivers on firm performance in the studied region, and whether outsourcing activities mediates this effect.

The literature states that a skilled labor force enhances the firm's performance in both large organizations and small to medium enterprises (SMEs) [4–6]. Nevertheless, large organizations have more resources available to invest in human capital, such as training and personnel development, which are essential for the improvement of the operations of organizations [7,8]. Consequently, this study aims to identify whether there are differences between SMEs and large organizations regarding the impact of human capital on the stated drivers. The differentiation is important as SME organizations represent more than 90% of the Colombian productive sector and generate more than 67% of the formal employment. SMEs also contribute 28% of the national GDP.

To test our assertions, we surveyed 435 respondents in job positions related to supply chain management (SCM) or operations in SMEs and large organizations (LOs) in the Valle del Cauca region in Colombia. The Colombian ministry of commerce defines the size of an organization based on the tax value unit (TVU), which is a value equivalent to the Colombian peso used to determine different tax obligations, such as the minimum amounts of withholding at source or penalties. Organizations with a TVU below 1.736.565 are considered SME. All other organizations are ranked as LO. The structural equation modeling (SEM) analyses show that human capital strategies are essential to both SMEs and LOs to ensure cost-effective, high-quality, and agile operations; that outsourcing is critical to facilitate agility; and that effective cost strategies positively affect firm performance, therefore confirming the literature reviewed in this article.

Our findings also hint that supply chain outsourcing is relatively scarce in the Valle del Cauca region, which seems to oppose widespread international trends. The studied region's strong reliance on in-house activities hinders service effectiveness and subsequent competitiveness in an increasingly global business context. Overall, the evidence from the data suggests that to gain a competitive edge in the global marketplace, local organizations should pay keen attention to the competitive dimensions addressed in this study.

The rest of the paper is organized as follows. Section 2 provides a conceptual framework used for the formulation of hypotheses and the presentation of the conceptual model of the relationship between the proposed variables, while section 3 presents the research method. The fourth section presents the most important results and findings. Section 5 shows the measurement items used. Finally, in section 6, the conclusions of the study, its limitations, and future lines of research are discussed.

## 2. Literature review

As firms cannot excel in all business dimensions, managers must identify key drivers, define competitive priorities, and formulate consequent strategies [1,3,9]. Especially in businesses that rely heavily on supply chain effectiveness, firm competitiveness requires standing out in cost, flexibility, quality, delivery speed, and reliability, as well as deploying a skilled workforce and promoting innovation. Such competitive drivers are key to developing organizational capabilities and competencies inherent to effective firm strategies [10,11].

## 2.1. Human capital

The academic interest in strategic human capital as a firm's competitive advantage source has grown consistently over the last decade. However, few studies have taken place in SMEs and none in countries like Colombia and the studied region (Valle del Cauca). Most extant research on these issues has taken place in large, private, Europe/US-based, multinational enterprises [12]. As a result, many of the established concepts explaining human capital strategies, largely defined in the context of such companies, might be tested and adjusted in the contexts of other regions and other organizational structures. Therefore, studies conducted in other regions, particularly in emerging economies, and that also compare SMEs to large organizations are valuable, as they add to the increasing body of knowledge obtained through researching the strategic role of human capital in achieving performance excellence.

Based on human capital theories [13], the resource-based view, and capabilities [14], from an economic-productive perspective, human capital can be defined as the set of resources, capabilities, and competencies that enable individuals to assume responsibilities at the organizational level. Its contribution to innovation and business performance has raised controversy. On the one hand, researchers such as Jabbour and de Sousa Jabbour [15] and Gorman et al [16] argue a direct impact of human capital on organizational results. On the other hand, researchers such as Huo et al. [17] and Crandall and Crandall [18] argue the human capital incidence is indirect. They suggest that human capital directly affects operational efficiency, which in turn, affects supply chain effectiveness and firm performance as a result.

The positive effects of human capital on the supply chain, operational efficiency, and business innovation are mediated by different human resources management practices [19]. Companies allowing an effective combination of alignment and flexibility between their human resources management and SCM should raise their performance [20,21]. In this sense, human resources practices can make important contributions to the firm's capability oriented towards the SCM from both an intra-organizational and inter-organizational approach gaining high performance. All this, by creating strategic partnerships, promoting intra-organizational and inter-organizational learning, building trust, and fostering synergy between different involved firms [20,21].

Specifically, training and teamwork [22,23], selective recruitment, variable payment, and, in particular, global thinking [24], improve the supply chain's sustained performance. Regarding employee incentives, Huo et al. [25] relate them positively with the supply chain internal integration, negatively with customer integration, and with no effect on supplier integration. The direct participation of employees in integration management is related to the three dimensions. Besides this, Liu, Combs [26] state that highly committed and trained human capital allows employees to handle complex tasks in the supply chain, both within and between companies. For Smith-Doerflein, Tracey and Tan [27], supply chain managers must ensure that human capital management practices match the company's needs. Likewise, they must ensure that employees have the skills and competencies for decision-making and effective solutions for diverse problems at a strategic, tactical and operational level [28].

To achieve operational effectiveness, organizations generally emphasize five dimensions: 1) cost, 2) quality, 3) reliability, 4) flexibility, and 5) speed [16,29]. In particular, for this research purpose, it is assumed that human capital is a critical driver of firm performance through the combined relationships of performance objectives such as cost, quality, and agility strategies. Likewise, based on Lepak and Shaw [30], Lepak, Takeuchi and Snell [31], and Lepak, Taylor, Tekleab, Marrone and Cohen [32], this study considers that human capital management effectiveness as a mediator of competitive performance is conditioned by the practices used in human capital management, the architecture and organizational infrastructure (related to the

supply chain in this case), and clarity about the required human capital features according to the expected strategic goals.

## 2.2. Agility

As firms are increasingly challenged to respond to ever-increasing changes in demand, supply chain managers are pressed by customers whose decisions are based on the "cost of time" [33]. Consumers value time and have tools available to them, based on the internet, that enable them to source from those suppliers that meet their cost and quality specifications faster than their competitors. Thus, success—and quite possibly, survival—in the marketplace often relies on being able to respond more rapidly and effectively to changing demands in product volume and variety [34].

Agility, understood as a comprehensive approach to responsiveness [35], concerns the extent to which upstream and downstream supply chain members are capable of reacting promptly and creatively to challenging demand and unforeseen marketplace variation [36]. The ability of a firm's supply chain to rapidly respond to market changes, creates customer value. Changes can affect fluctuation in the portfolio of products, the required volume, or delivery sequence or timing, therefore requiring, respectively, mix responsiveness, volume responsiveness, and delivery responsiveness. The meaning of short and medium-term are relative and context dependent. Changes in some industries—such as electronics, where product life cycles can be as short as just a couple months—are much shorter-termed than in other industries—such as automotive, where models remain in production for six years on average [37–39].

Past research has shown that firms that are more agile than their competitors in different, key aspects of their operations tend to outperform them consistently [40–42]. By developing an agile supply chain, firms can respond better to changing market situations and incur lower inventory holding costs, thus yielding higher returns on their capital.

As agility is constantly a pursued firm's key performance objective, lead time reduction has arisen as a prominent practice in manufacturing and supply chain strategy [41,43,44]. Reducing lead time results in better profitability, lower costs, better inventory rotation, more efficient scheduling, and better service [45]. Firms that promise and provide faster delivery times can charge a premium price and enjoy higher customer demand [46]. The lead-time reduction can assist organizations in reducing inventory costs [47,48]. With longer lead times, schedules and assets must be frozen over a longer time. This, in turn, raises the likelihood of incorrect demand forecasts and calls for higher safety stocks to account for demand variability and prevent stock-out problems and their associated costs during replenishment.

Agility and the associated lead-time reduction greatly benefit from the information shared along the supply chain. Sharing information reduces uncertainty, time, and ordering process costs. Conversely, shorter lead-times allow firms to focus on and optimize demand information sharing. In contrast, longer lead-times force firms to spend efforts and resources to integrate their planning and forecasting information with supply chain associates [49].

Firms should realize that adversarial relationships prevent all players from exploiting opportunities for greater agility [36]. The high levels of information sharing and connectivity present in long-term and close relationships enable the notion of agility. Collaborative relationships or alliances are not always workable, so firms should carefully examine how to develop supply chain agility in less cooperative, not-so-close relationships [50].

As human capital receives wider attention with increasing globalization, most economies emphasize the improvement of human capital towards fast-tracking their growth. Hence, human capital development is one of the critical approaches to becoming more competitive in

the international arena [51,52]. Therefore, the skills of both managers and workers make the organization effectively implement strategic initiatives that boost competitive advantages [53].

Based on a survey answered by 201 supply chain managers in the manufacturing sector, Jin, Hopkins [54] found direct links between human capital, the flexibility of the operations, and the production time to gain a competitive advantage. Consequently, based on Kraatz and Zajac [55] and Jin et al [54], this study contends that human capital strategies are essential to ensure agile, cost-effective, and high-quality operations. Thus, it is hypothesized that:

*Hypothesis 1*: *Human capital is a positive predictor of a firm's agility strategy.*

## 2.3. Quality

The extant literature proposes a myriad of definitions for quality; different meanings might be appropriate under different contexts. From a manufacturing perspective, quality refers to fulfilling standards [56–58], a product-oriented approach that ensures that products are made well from the beginning. From a marketing perspective, quality relates to meeting or exceeding customer's expectations [59,60], a challenging definition given that identifying such expectations is a complex task. From a more strategic view, quality has been well recognized by decision-makers and quality professionals as an initiative to total quality and change management approaches [61,62]. Quality is therefore a vital aspect in the development of management practices [63]. Total quality management (TQM) is a key practice for improving effectiveness, flexibility, and business competitiveness to meet customers' requirements leading to a competitive edge [64]. Furthermore, TQM has been asserted as a source of excellence achievement, building a right-first-time attitude, attaining efficient tactical solutions, satisfying beyond expectations the clients and suppliers, etc., and above all, as a source of boosting organizational performance through sustained improvement in supply chain firms' activities [65].

From the supply chain perspective, quality management can be recognized when the supply chain design across every organization provides quality products and services according to clients' expectations [66]. The quality improvement of all supply chain processes results in reductions in cost, resource utilization improvement, and process efficiency improvement [67]. Research has been conducted on how quality practices can be used to improve the entire supply chain and address problems within the supply network [68].

It has been suggested that more work is needed for a better knowledge of continuous improvement practices in the supply chain context and the linkage between quality approaches and overall organizational performance. Consequently, standards for the implementation of quality practices in the supply chain are suggested for future research that can be of use for firms. For example, Dowlatshahi [68] and Lin and Gibson [69] performed exploratory studies on the role quality management plays in integrated supply chain performance. They propose that more work is needed on the reason why quality practices strongly impact an integrated supply chain. On the other hand, few studies have been conducted on the association between quality and firm performance mediated by strategic practices such as outsourcing.

Aryee, Walumbwa [70] reveal that high-performance work systems are related to human capital and the quality of the service at the individual level. Furthermore, Marimuthu, Arokiasamy [51] maintain that human capital has a direct impact on firm performance from various critical perspectives. Also, Snell and Dean [52] claimed that quality practices are positively related to human resource practices and to the comprehensiveness of training for operations employees to achieve higher levels of quality; therefore, it is hypothesized that:

*Hypothesis 2*: *Human capital is a positive predictor of a firm's quality strategy*

## 2.4. Costs

Costs are the financial expenses incurred in engaging in business which is essential to evaluate the performance of a firm. Supply chain cost is defined as all relevant expenses of the operations of the supply chain of the company or organization in question. An effective SCM will reduce the operational costs of a firm [71,72]. Costs can be measured in terms of the supply chain as a whole [73], each supply chain process [74], or the logistical costs only [75]. Analysis of the supply chain costs can be performed in different ways. Different kinds of cost groupings can be found in the literature, including transportation, production, inventory, packaging, materials holding, and order processing [76].

The need to measure and control supply chain costs is increasingly relevant as firms grow in size and scope [77]. Supply chain partners should, therefore, actively seek cost-saving opportunities along the supply channels by gaining a better understanding of the linkages between different value-adding activities [78].

This research is particularly interested in two important, and often overlooked, cost elements: transaction costs and switching costs [79,80]. Information becomes available to supply chain participants at a cost. Firms incur costs when searching for information and safeguarding against opportunistic behavior from other participants. When information is shared more frequently, searching costs and information-sharing transaction costs tend to decrease [81], which in turn decreases the overall cost of products and services [82]. Furthermore, asset specificity tends to increase as information is shared more freely and more frequently. Interestingly, a reduction in transaction costs comes as a trade-off to an increase in switching costs, given that it becomes more costly for supply chain partners to seek new partners in the marketplace.

A study by Birdi, Clegg [83] found that empowerment of the human capital, extensive training, and teamwork have a predictive role in the success of total quality and just-in-time initiatives and, therefore, the achievement of better performance and reduction of the transaction costs. Additionally, the integration of human capital practices in a lean manufacturing system and the customer-supplier relationships are necessary for the delivery of desired standards at a minimum cost [84]. Thus, this study hypothesizes that:

*Hypothesis 3*: *Human capital is a positive predictor of a firm's cost strategy*

## 2.5 Outsourcing

The outsourcing of elements of supply chain processes is now an important component of the operationalization of an organization's competitive strategy [85,86]. Outsourcing is commonly defined as the transference of activities from an internal operation processing point of view to external professional third parties [87]. Outsourcing of activities has been historically linked to cost-saving, third-party logistics (3PL) procurement strategies in manufacturing firms [88]. With increasing frequency, though, outsourcing is becoming an important strategic option for all kinds of organizations, and its benefits transcend the usual cost reductions associated with economies of scope and productivity by specialization [89].

Logistics outsourcing is one of the different forms of outsourcing that has attracted the attention of organizations, academics, and researchers in recent years. Its relation to the organizational bottom line serves the following purposes: leadership, accessing high-quality processes, harmonizing with technology alterations, and downsizing, which contribute to providing a competitive edge Erturgut [90].

The increment of global service transactions that is reflected in the growth of outsourcing and offshoring provides an opportunity to understand the role of outsourcing and offshoring

and the involvement of human capital in the quest to promote competitiveness [91,92]. Contractor and Mudambi [91] studied the effect of human capital on the exports of both goods and services. However, the results indicated that human capital was not significantly more important for services exports than for goods exports.

Outsourcing is, in some cases, an opportunity to increase the skill premium between foreign countries such as is the case of the US and Mexico that base their relationship on the North American Free Trade Agreement [93]. Interestingly, highly talented workers in Austria are losing benefits due to outsourcing, whereas in Poland, outsourcing is providing benefits [92]. Additionally, it is known that the impact of human capital investment has affected emerging economies like Asia significantly more than developed countries [91]. Thus, given the discrepancies in the impact of human capital on outsourcing initiatives, this study hypothesizes that:

*Hypothesis 4*: *Human capital is a positive predictor of a firm's outsourcing strategy*

Hsu [94] found that different strategies related to logistics outsourcing can contribute to the reduction of fixed costs, thereby increasing flexibility. At the same time, it may be expected that logistics outsourcing will allow organizations to concentrate their activities on their business core, investment reduction, and the improvement of the quality of services.

Consequently, Deepen [95] acknowledges that logistics service providers (LSPs) are more efficient than logistics service users (LSUs). Deepen [95] also asserts that cost reduction can be seen as the greatest benefit derived from the help given by LSPs to LSUs. Consequently, the LSU cost position is affected by the reduction of capital investment requirements due to the LSP outsourcing processes. The connection between LSPs and LSUs is reinforced through valuable help from the former so the latter avoid any unnecessary investment in both the management of the workforce and their supply chain.

The win-win relationship between LSPs and LSUs promotes, through appropriate supply chain management, benefits such as cost reductions, better flexibility, and an increased level of responsiveness to the customers' requirements. Several researchers have conducted studies to identify the main reasons for outsourcing. However, even though cost reduction is one of these, it is not always the main motivation [96]. Whether the organization engages in customer-related versus strategic activities will define the reasons for outsourcing [97].

Different criteria have been transcendental to outsourcing since its first implementation. Alkhatib et al. [98] conducted a comprehensive literature review and identified cost, quality, flexibility, services, financial measures, sustainability, and delivery as the most relevant and important criteria since the mid-1960s, and these account for 76.83% of the criteria used during 2008–2013 for outsourcing.

Table 1 shows the relative importance of the aforementioned criteria in different periods from the 1960s to the present.

The same study categorized the criteria into three main dimensions: Performance (financial, customer, and operational), Resources (tangible and intangible), and Services.

Most Fortune 500 firms base their operations on some form of outsourcing, a pattern that is growing throughout industries and regions around the world. Such popularity

**Table 1. Criteria relative importance (1966–present).**

| Period | High level of importance | Medium level of importance |
|---|---|---|
| 1966–1990 | Cost and delivery | Quality |
| 1990–2008 | Quality | Cost and delivery |
| 2008–present | Cost and price | Quality |

notwithstanding, there are still untapped areas where outsourcing has to prove its full potential, such as SMEs, outsourcing in emerging economies, and exploiting leveraging capability development in outsourcing strategies. In a globalized marketplace, SMEs are increasingly pressed to proactively seek and develop strategic partnerships in the supply chain [99]. Outsourcing can be a catalyst for innovation, quality, and supply chain flexibility, and thus contribute to the achievement of competitive advantage, provided that cross-functional in-house activities within the firm are effectively linked to externally executed processes [100]. Therefore, this study hypothesizes that:

*Hypothesis 5*: *A firm's outsourcing strategy is a positive predictor of its cost strategy*

*Hypothesis 6*: *A firm's outsourcing strategy is a positive predictor of its quality strategy*

*Hypothesis 7*: *A firm's outsourcing strategy is a positive predictor of its agility strategy*

## 2.6. Business performance

Performance at the operational level relates to the degree to which organizations make progress towards key metrics such as cost-efficiency, quality, reliability, and flexibility [101,102]. Operational performance attainment encompasses building key capabilities with which organizations develop strengths that enable them to sustainably outperform challenges in their marketplace [103]. Furthermore, it is often asserted that frequent implementation of quality management and agility programs improve the achievement of operational objectives, including cost-effective performance, enhanced quality, shorter lead times, and flexible processes. Waste management, via process defects detection and fixing, also benefits from the implementation of quality programs [104].

Business performance relates to the outcomes of certain coordinated activities through the utilization of company resources to ensure the successful execution of business strategy for the fulfillment of organizational goals. Business strategy impacts the level of relationship strength between particular technology policies and firm performance [38].

At the business level, business performance relates to the utilization of firms' resources to perform synchronized functions that ensure meeting shareholder's expectations and achieving profit maximization [105]. Business performance is often conceptualized through the execution of two dimensions: market performance (e.g., market power, market efficiency, market volatility) and financial performance (e.g., profitability, liquidity, solvency, the margin on sales) [101].

As shareholder's expectations are very dynamic, the measurement of business performance is expected to become more complex. The latter suggests several realistic scenarios that prove the complex performance-related dilemmas firms are confronted with: 1) Shareholder elevated returns vs. detrimental human resource management; 2) Organization as the "Best Employer" vs. weak financial performance; 3) Company recognized by self-imposed environmental standards vs. low efficiency of the manufacturing process, increased costs, low employee morale, and job cuts.

Agility, quality, and cost are key drivers of firm performance [106,107]. Being agile allows the organization to respond in a faster way to consumer requirements and to provide products or services of higher quality than the competitors. Pursuing agility within the organization means involving people in decision-making, adequate training facilities, and a motivating reward program [108]. Also, an organization with productive human capital can deliver products and services of higher quality than competitors, thus generating a positive impact on performance [109]. The cost has an impact on performance. A cost orientation strategy, based on

human capital, can create a superior firm performance [110]. Consequently, an adequate human capital structure enhances the capacity of an organization to compete in agility, quality, and cost strategies and, therefore, impact positively on performance.

Based on the literature reviewed, this study contends that human capital strategies are essential to ensure cost-effective, high quality, and agile operations, that outsourcing is important to facilitate the effectiveness of such drivers, and that effective deployment of these key drivers positively impacts firm performance and competitiveness. Therefore, this study hypothesizes that:

*Hypothesis 8*: *A firm's human capital is a positive predictor of the firm's performance*

*Hypothesis 9*: *A firm's cost strategy is a positive predictor of the firm performance*

*H10*: *A firm's quality strategy is a positive predictor of the firm's performance*

*H11*: *A firm's agility strategy is a predictor of firm performance*

*H12*: *A firm's outsourcing strategy is a predictor of firm performance*

Fig 1 illustrates the hypothesized relationships and the corresponding SEM model to be tested in this study.

## 3. Research methods

This study tested several hypotheses through an online survey submitted to multi-sectorial firms in the Valle del Cauca agro-industrial region, in Colombia (see Table 2). Ferrer et al [111] conducted a similar study in Saudi Arabia that considers the impact of human capital on operational strategies such as cost, quality, flexibility, and outsourcing initiatives. However, a

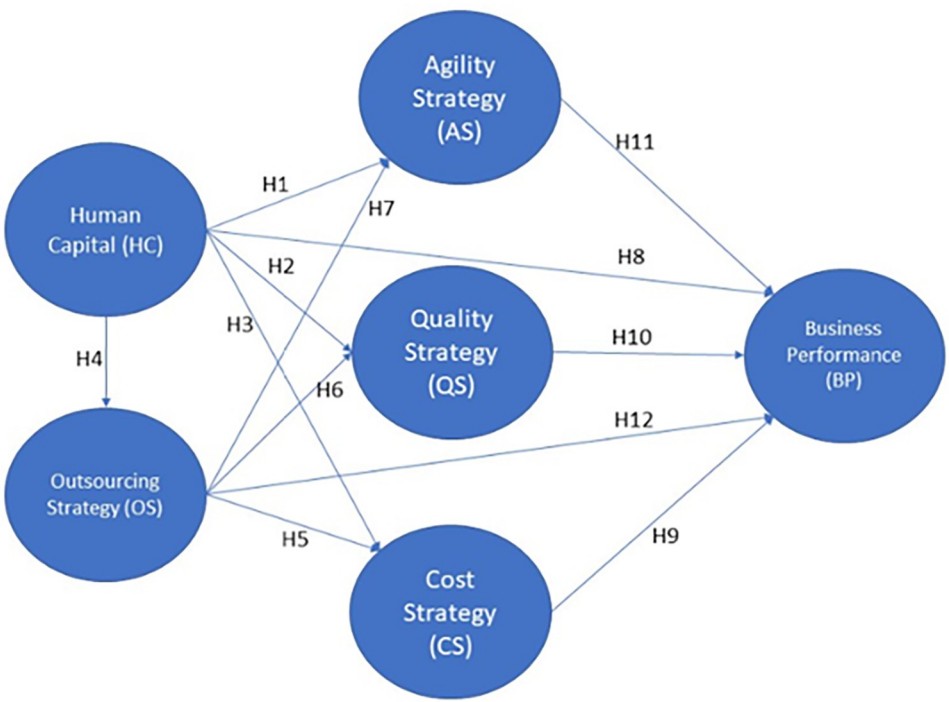

**Fig 1. Research model with hypotheses.**

**Table 2. Respondents by sector and professional role.**

| Industrial Sector | Frequency | % | Professional Role | Frequency | % |
|---|---|---|---|---|---|
| Manufacturing | 174 | 43.1% | Line Managers | 102 | 25.2% |
| Aerospatial Industry | 125 | 30.9% | Logistics | 73 | 18.1% |
| Energy | 27 | 6.7% | Project Management | 54 | 13.4% |
| Agriculture | 16 | 4.0% | Operations Manager | 46 | 11.4% |
| Technology | 16 | 4.0% | Production | 34 | 8.4% |
| Retail | 13 | 3.2% | Accounting | 25 | 6.2% |
| Health care | 11 | 2.7% | Executive Manager | 22 | 5.4% |
| Construction | 10 | 2.5% | Finance | 19 | 4.7% |
| Government | 10 | 2.5% | Other role | 19 | 4.7% |
| Other sector | 2 | 0.5% | Marketing | 10 | 2.5% |

careful literature review found no precedents to this kind of study comparing SMEs and large organizations in the region and the factors selected for this research. Employing a confirmatory–correlational design, we set up this study to explain, quantify, and determine the relationship between the variables of interest, and to identify possible underlying causes that account for observed patterns [112,113]. Hair et al [114] provided the guidelines to test the model (Fig 1) based on a set of appropriate constructs.

Statements related to the operationalization of the various constructs were evaluated based on a five-point Likert-type scale (Strongly Disagree—Strongly Agree) representing the perception of the respondents according to the different questions. The first section was designed to profile the respondents. The second section included statements related to supply chain practices from Ferrer [115], who adapted measures from the characteristics of the relationships scales developed by Knemeyer and Murphy [116] combined with Ellram [117] who aimed at classifying each service contractor's relationship with supply chain partners into a particular stage of evolution. The sections on innovation, quality, cost, flexibility, and speed had 31 statements adapted from the scale operationalized in the model developed by Kroes [85]. Supply chain performance-related scale items were also adapted from Kroes [85]. Scale items related to outsourcing inhibitors were drawn from Hung Lau and Zhang [118] and Kroes [85]. Furthermore, skilled-worker-related scale items were developed for this study based on factors extracted from Smith [119] and Hoecht and Trott [120]. Appendix A shows the measurement items.

Given the advantages of online surveys [121], we invited potential respondents to participate by email, with a link to access the questionnaire in Microsoft Forms, from which the data can be transferred directly to SPSS. In total, 435 participants submitted their responses, for a 72.5% response rate. After a careful review, 31 surveys were discarded due to inconsistencies, for a final count of 404 usable surveys (67.3% effective response rate). The structural equation model's variables were built by using the average mean values of the statements' ratings. This methodology fits the appropriateness for our particular research and the analysis of latent variables and their cross relationships, as well as the fitness between the required sample and the data collected [122]. The collected surveys were split into two distinct groups, small to medium enterprises (SME) (N = 167) and large organizations (LO) (N = 237) to gain an understanding of how competitive priorities relate to firm size.

The study was approved by the Ethics Committee at Colegio de Estudios Superiores de Administración (CESA): Approval number 006, dated 25-04-2022. Before filling out the questionnaire, the respondents were informed that all information provided would be treated in strictest confidence, that the responses would be aggregated and used for research purposes,

that by completing this survey they were giving their consent to such use, and that they might withdraw at any time. Ethical procedures were followed by the researchers during this research project.

Table 2 shows the demographics for this study by the respondent sector and area of responsibility. The data collected indicates that 43% of the respondents were manufacturing organizations, particularly linked to the ministry of national defense where the study was developed in partnership with a local university. Organizations linked to the aerospace industry represent 30.9% of the responses, and the rest of the responses were collected from organizations associated with the government, such as providers of services to the military industry. Another important factor about the sample is that the respondents have jobs related to operational activities.

We used SPSS [123] and AMOS software to analyze the data. The analyses included confirmation of the conceptualized model shown in Fig 1. The following steps summarize the statistical analysis performed to estimate the model predictiveness as well as its indices.

1. Confirmatory factor analysis (CFA) with the following settings:

   - Discrepancy: Asymptotically distribution-free
     Covariances supplied as input

   - Unbiased
     Covariances to be analyzed

   - Maximum likelihood

2. Factor loading estimation (assessment of the relationship between the observed and continuous latent variables)

3. Internal consistency (Cronbach's alpha coefficient shown in Table 3) [124,125].

4. Testing of construct validity (CFA)

Table 4 summarizes the results for both models for SMEs and LOs. The resulting indices (SME: CMIN/DF = 2.945, probability = 0.000; and LO: CMIN/DF = 4.168, probability = 0.000) fall within the accepted criterion of CMIN/DF$\leq$5.0 [126]. Also, a comparative fit index (CFI<0.9) is usually accepted as supportive of the model [127], which is the case in the hypothesized model (SME: CFI = 0.915; LO: CFI = 0.901). Table 3 shows the high degree of fitness of the proposed model concerning the null model.

## 4. Results and discussion

Table 5 and Figs 2 and 3 summarize the SEM's results on the relationships described in the model in Fig 1. For SME, there is a strong and positive relationship between HC and all three

**Table 3. Cronbach's alpha for small–medium and large organizations.**

| | Small to Medium Enterprises (SME) | | Large Organizations (LO) | | The customary cut-off level for basic research |
|---|---|---|---|---|---|
| Variable | Total Items | Alpha (α) | Total Items | Alpha (α) | |
| Human capital (HC) | 3 | 0.933 | 4 | 0.919 | 0.70 |
| Cost Strategy | 3 | 0.964 | 4 | 0.944 | 0.70 |
| Quality Strategy | 4 | 0.922 | 3 | 0.938 | 0.70 |
| Agility Strategy | 9 | 0.825 | 4 | 0.933 | 0.70 |
| Outsourcing | 3 | 0.894 | 4 | 0.893 | 0.70 |
| Business Performance | 4 | 0.852 | 4 | 0.897 | 0.70 |

**Table 4. Baseline comparison.**

| Model | Small to Medium Enterprises (SME) | | | | | Large Organizations (LO) | | | | |
|---|---|---|---|---|---|---|---|---|---|---|
| | NFI<br>Delta 1 | RFI<br>Rho 1 | IFI<br>Delta 2 | TLI<br>Delta 2 | CFI | NFI<br>Delta 1 | RFI<br>Rho 1 | IFI<br>Delta 2 | TLI<br>Delta 2 | CFI |
| Default | .880 | .854 | .916 | .897 | .915 | .839 | .817 | .896 | .875 | .904 |
| Saturated | 1.000 | | 1.000 | | 1.000 | 1.000 | | 1.000 | | 1.000 |
| Independent | .000 | .000 | .000 | .000 | .000 | .000 | .000 | .000 | .000 | .000 |

types of strategy (AS: b = 0.76, p<0.001; QS: b = 0.41, p<0.001; and CS: b = 0.35, p<0.001), thereby supporting hypotheses H1, H2, and H3.

Findings for LO were similar, also indicating a strong and positive relationship between HC and all three types of strategy (AS: b = 0.67, p<0.001; QS: b = 0.55, p<0.001; and CS: b = 0.30, p<0.001), thereby supporting hypotheses H1, H2, and H3. Concurring results for SME and LO endorse the importance of implementing adequate human resources practices to ensure agility, quality, and cost management, regardless of firm size.

Also, H4 is supported for both SME (b = 0.26, p<0.001.) and LO (b = 0.19, p<0.01.). This result enhances the importance of HC in creating an adequate network to foster outsourcing strategies. Hypothesis H5, that there is a positive relationship between a firm's cost strategy and its outsourcing strategy, was rejected for SME (b = 0.17, n.s.), but not for LO (b = 0.19, p<0.01). These results suggest that SMEs do not seek cost efficiency by outsourcing activities, unlike LOs. Hypothesis H6, which suggests that firms that prioritize quality strategies will tend to implement outsourcing initiatives, was also rejected for SME, but not LO (b = 0.04, n.s. and (b = 0.15, p<0.01, respectively).

Interestingly, hypothesis H7, that there is a positive relationship between a firm's agility strategy and outsourcing, was supported for SMEs (b = 0.22, p<0.001), but not for LO (b = -0.04, n.s.). It appears that whereas outsourcing is a source of agility for SMEs to satisfy customers' requirements, this is not the case for LOs. It could also be explained by the fact that to compete more effectively against stronger and wealthier rivals with a larger market share, SMEs are pressed to be more flexible and timely, but the agility gained by SMEs is detrimental to the quality and cost strategies. This behavior is contrary to that of LOs, which indicates that the competitive priorities depend on the size of the organization.

**Table 5. Regression weights.**

| | | | Small to Medium Enterprises (SME) | | | | Large Organizations (LO) | | | |
|---|---|---|---|---|---|---|---|---|---|---|
| | | | Estimate | S.E. | C.R. | P | Estimate | S.E. | C.R. | P |
| OS | <— | HC | .325 | .097 | 3.334 | *** | .261 | .094 | 2.775 | .006 |
| AS | <— | OS | .154 | .039 | 3.953 | *** | -.027 | .037 | -.738 | .461 |
| QS | <— | OS | .024 | .043 | .565 | .572 | .112 | .043 | 2.593 | .010 |
| CS | <— | OS | .131 | .060 | 2.197 | .028 | .150 | .051 | 2.962 | .003 |
| AS | <— | HC | .672 | .070 | 9.594 | *** | .635 | .060 | 10.628 | *** |
| QS | <— | HC | .290 | .058 | 5.035 | *** | .588 | .064 | 9.237 | *** |
| CS | <— | HC | .338 | .077 | 4.373 | *** | .327 | .072 | 4.542 | *** |
| BP | <— | OS | .045 | .069 | .644 | .520 | .123 | .063 | 1.965 | .049 |
| BP | <— | HC | -.907 | .175 | -5.196 | *** | -.288 | .137 | -2.102 | .036 |
| BP | <— | AS | .752 | .197 | 3.809 | *** | .047 | .120 | .395 | .693 |
| BP | <— | QS | .201 | .114 | 1.762 | .078 | .489 | .099 | 4.922 | *** |
| BP | <— | CS | .197 | .085 | 2.325 | .020 | .291 | .084 | 3.470 | *** |

OS: Outsourcing; HC: Human Capital; AS: Agility Strategy; QS: Quality Strategy; CS: Cost Strategy; BP: Business Performance.

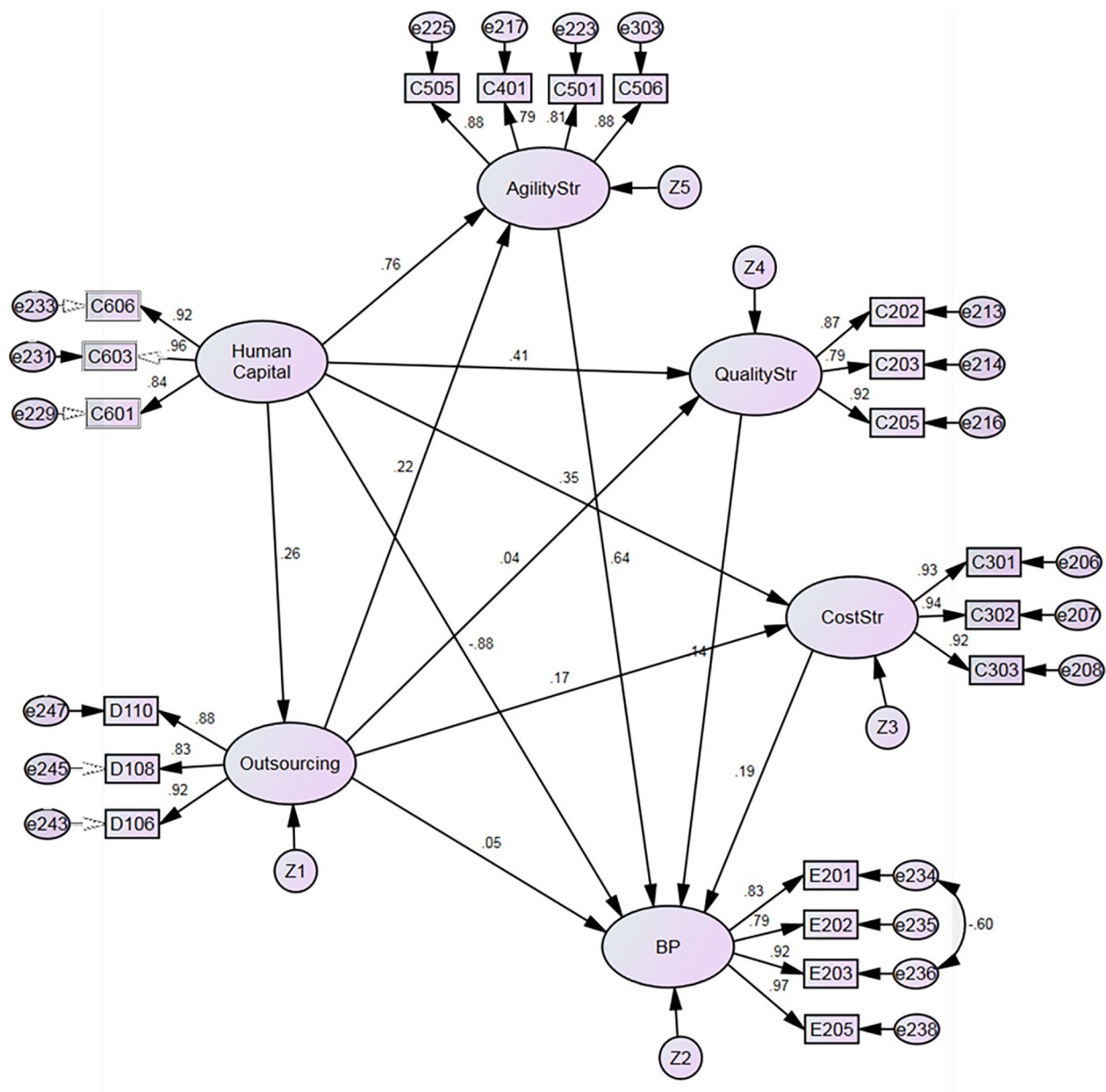

**Fig 2. Structural model results for SME.**

Human capital has a negative and significant impact on business performance for SME (b = -0.88, p<0.001) and no impact for LO (b = -0.288, n.s), and thus, does not support H8. This fact suggests that for both SMEs and LOs there is a gap between the strategic role of human capital and the performance of the organization, influenced by the lack of a strategic role of HC on the supply chain. For SMEs, the fact that the impact of HC on BP is negative and significant demonstrates that SMEs are less formalized, have less financial muscle than LOs, and also, they are not applying adequate HC strategies. The HC department plays an important role in employee development, improving quality of life and income, increasing knowledge and skills aimed to improve production, facts that are essential for the economic growth of the organization and the region. The HC department plays a key role in achieving better

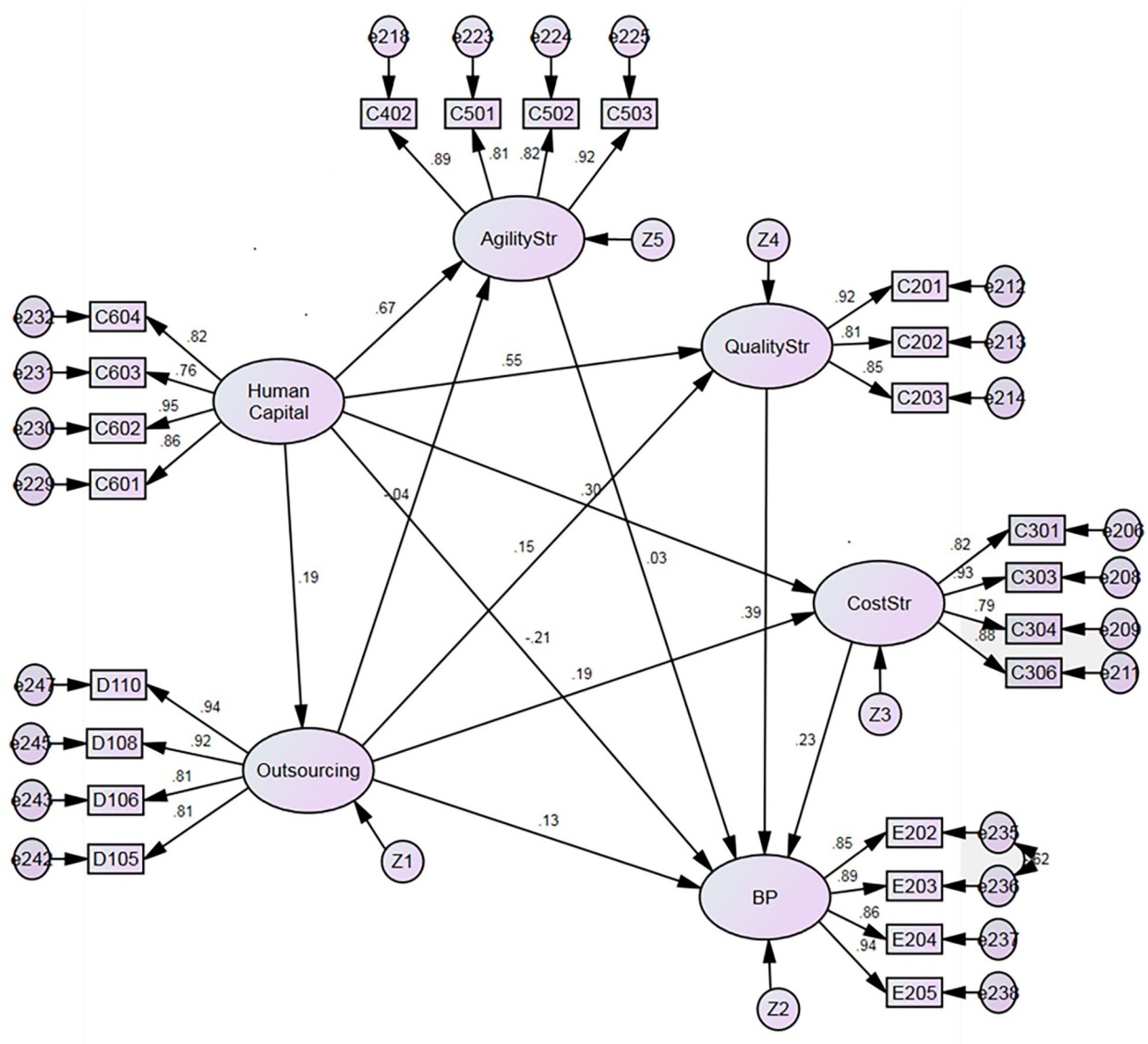

**Fig 3. Structural model results for LO.**

performance by hiring adequate personnel, providing the necessary training, and also designing policies to solve any possible disagreement between coworkers that could harm business performance.

This study found a positive relationship between cost strategy and firm performance, for LO (b = 0.23, p<0.001), but not for SME (b = 0.19, n.s.), thereby supporting hypothesis H9 for LO, but not for SME. Hypothesis H10, that there is a positive relationship between quality strategy and firm performance, was rejected for SME (b = 0.14, n.s.) and supported for LO (b = 0.390, p<0.001). Hypothesis H11, on the relationship between agility and performance, was rejected for LO (b = 0.03, n.s.), but not SME (b = 0.64, p<0.001.). These results may be explained by the focus of SMEs on the agility to respond to customer needs, and customers valuing such efforts. LOs, with their more inflexible routines and work practices, rely primarily on cost and quality (as shown in H9 and H10) to enhance business performance and gain a competitive advantage.

**Table 6. Hypothesis tests.**

| Hypotheses | Predictor variable | Dependent variable | Results | SME | LO |
|---|---|---|---|---|---|
| H1 | Human Capital | Agility Strategy | SME: b = 0.76, p <0.001<br>LO: b = 0.67, p <0.001 | Supported | Supported |
| H2 | Human Capital | Quality Strategy | SME: b = 0.41, p <0.001<br>LO: b = 0.55, p <0.001 | Supported | Supported |
| H3 | Human Capital | Cost Strategy | SME: b = 0.35, p <0.001<br>LO: b = 0.30, p <0.001 | Supported | Supported |
| H4 | Human Capital | Outsourcing | SME: b.26, p <0.001.<br>LO: b = 0.19, p <0.01 | Supported | Supported |
| H5 | Outsourcing | Cost Strategy | SME: b = 17, n.s.<br>LO: b = 0.19, p <0.01 | Rejected | Supported |
| H6 | Outsourcing | Quality Strategy | SME: b = -0.04, n.s.<br>LO: b = 0.15, p < 0.01. | Rejected | Supported |
| H7 | Outsourcing | Agility Strategy | SME: b = 0.22, p <0.001<br>LO: b = -0.04, n.s. | Supported | Rejected |
| H8 | Human Capital | Business Performance | SME: b = -.88, p <0.001<br>LO: b = —0.21, n.s. | Rejected | Rejected |
| H9 | Cost Strategy | Business Performance | SME: b = 0.19, n.s.<br>LO: b = 0.23, p <0.001 | Rejected | Supported |
| H10 | Quality Strategy | Business Performance | SME: b = 0.14, n.s.<br>LO: b = 0.39, p <0.001 | Rejected | Supported |
| H11 | Agility Strategy | Business Performance | SME: b = -0.64, p <0.001<br>LO: b = 0.03, n.s. | Supported | Rejected |
| H12 | Outsourcing | Business Performance | SME: b = -0.05, n.s.<br>LO: b = -0.13, n.s. | Rejected | Rejected |

Another important finding inferred from the results is the mediating role that the strategic competitive drivers, quality, cost, and agility have between HC and BP. Agility is a clear mediator between HC and BP for SMEs, and cost and quality are mediators between HC and BP for LOs. This phenomenon confirms the key role played by competitive drivers in any organization as they connect employees with the strategic performance objectives to gain a competitive advantage.

Finally, hypothesis H12, which suggests that firms will improve performance through the use of outsourcing practices, was rejected for both SME (b = 0.05, n.s.) and LO (b = 0.13, n.s.). Table 6, together with Figs 2 and 3, summarizes the SEM hypotheses' tests.

## 5. Conclusion

This article has defined the human capital and key competitive driver effects on supply chain operations and firm performance in the Valle del Cauca region in Colombia. The findings suggest human capital is essential for agility, quality, and cost-effectiveness strategies, which, in turn, impact positively on firm performance. Furthermore, results are valid for both SMEs and LOs, however, for SMEs, the human capital impact on organizational performance is only through agility strategies, whereas for LOs, it is through cost and quality strategies.

Aligned with much of the extant literature on human resources, the human capital impact on firm performance is therefore indirect, and results are rather attained through effective quality, cost, and agility strategies. Consequently, the human capital effects on SME and LO performance will not depend on human capital features only but also on the company's infrastructure and architecture in which it operates as a driver.

However, despite the human capital and competitive drivers' importance, the relationship between them and performance is not as important nor as strong for SMEs as it is for LOs in

the studied region. While the cost and quality strategies are important for LOs to achieve competitiveness (superior business performance), agility strategies are related to the company's performance only for SMEs. The cost of implementing TQM initiatives (and other similar systems related to quality) could explain the former, while the need to be flexible and timely to compete against stronger rivals could explain the latter.

Another interesting inference from the results is the companies' preferences towards outsourcing strategies as a driver of business performance. Contrary to the trends of international companies, the studied firms seem to ignore outsourcing practices as a source of competitiveness. The relationships between outsourcing strategies and cost, quality, and agility strategies are not as important as they should be. The use of outsourcing as a way to achieve agility is not clear for LOs, but it is important for SMEs, which characterize themselves as flexible and agile due to their level of specialization and compromise. SMEs develop their outsourcing activities based on their motivation to overcome the lack of access to knowledge, technological and capital resources that LOs enjoy.

It is important to realize that the outsourcing benefits for the companies studied still focus on re-definition and re-organization of operations to continue strengthening and taking advantage of what they already do well through speed and efficiency. It is possible that in Colombian SMEs, outsourcing decisions are made based on the personal interests of the management team. Furthermore, it is inferred that SME outsourcing decisions can be made based on human resources needs, specific activities interests, or convenience, instead of considering how to support the organization's entire strategy, based on the fact that they cannot compete through cost or quality.

Based on the foregoing, it could be argued that Colombian SMEs should establish key processes to begin strategic mechanisms and routines to decide on which outsourcing practices or activities to engage in because they lack competitive advantage compared to LOs. However, the absence of such mechanisms makes it difficult to determine what functions must be employed to guarantee reliable support for the organization.

The results suggest that local companies should strengthen their alliances with logistics partners (3PL) to be competitive internationally. The benefits generally associated with 3PL practices (for example, service level control, systems integration, access to a variety of specialized resources, and cost reduction) are important for competitiveness in a globalized market. Such practices, however, do not seem to be usual in the analyzed region. It should be noted that large internationally recognized 3PL providers are essentially absent from the studied region, which also explains our results. This situation is probably a consequence of infrastructure deficiencies, with very modest multimodal connectivity. The development of specialized storage-free areas and better transport and communications infrastructure is certainly necessary to encourage stronger outsourcing activities, greater economic efficiency, and long-term alliances associated with it.

On the other hand, organizational culture, national or regional culture, limitations in financial or human resources, and other factors, such as trust, could hamper more proactive outsourcing strategies, especially among SMEs. Additionally, the study shows that the effective implementation of quality, cost, and agility strategies combined synergistically with outsourcing strategies improves the company's performance, perhaps compensating for the lack of other resources or capabilities.

However, it is clear human capital and the key drivers addressed in this study are not the only ones that affect performance. Beyond deepening the nuances of the relationships confirmed in this document, future studies could also explore cultural or regional specificities that better explain the role played by human capital and competitive drivers (or lack thereof) between the companies in the Valle del Cauca region. For example, companies may lack the

ability to obtain quality information related to 3PL's best practices and, therefore, ignore the strategic value of an effective partnership in the supply chain, the risk reduction potential, and the shared risk implicit in smart outsourcing strategies. They also ignore the possibilities of achieving the release of resources through 3PL to invest them in other valuable fields related to the core business.

Finally, the contribution of this study to the literature is threefold. Exploring the human capital influence on the relationships between the predictors of quality, costs, and agility, and the results of performance through outsourcing, not only contributes to a better understanding of human capital and the role of key competitive drivers but also highlights the regional specificities that could help to bring these issues to the consideration of managers for the formulation of more effective strategies. Additionally, alternative research methodologies are contributed by proposing a quantitative approach that helps fill the numerical gap common to many studies on competitiveness, particularly in developing countries.

## 6. Appendix A. Measurement items

In this section, please, state how critical the following competitive drivers are to your company (from Absolutely critical to not critical at all)

### Quality

High conformance of final products to design specifications
Excellent product performance
Superior product reliability
Fast resolution of customer complaints / inquiries
Offer consistent quality High product durability

### Cost

Maximize capacity utilizations
Increase labor productivity
Reduce production costs
Minimize inventory costs
Reduce distribution costs

### Flexibility

Make fast capacity adjustments
Make deliveries adjustments to fulfil customer requirements
Offer large number of product features
Offer product customization
Offer a large extent of product assortment
Adjust production processes to allow new product manufacturing

### Speed

Minimize set up times
Minimize delivery times
Minimize lead times
Fast introduction of new products
Minimize product development cycle time
Increase inventory rotation

### Skilled workforce

Well trained logistics workers
Resourceful front-line logistics workers
Experienced supply chain managers
Customer oriented logistics workers
Readily available outsourced logistics workers
A good pool of expert logistics workers

### In this section, please, state to what extent you agree or disagree with following reasons for not outsourcing logistics services

Control over the outsourced function (s) would diminish
Service level commitments would not be realized
Cost reductions would not be experienced
Local capabilities of 3PLs need improvement
Logistics too important to consider outsourcing
Corporate philosophy excludes the use of outsourced logistics providers
Issues relating to security of shipment
We have more logistics expertise than most 3PL local providers
Inability of 3PL providers to form meaningful and trusting relationships
Logistics is a core competency at our company
Our company previously outsourced logistics, and chose not to continue
Too difficult to integrate our IT systems with 3PL's systems
Other (Provide details)

### Business Performance

Our company has increased its profit margin
Our company has sustained competitive prices for its products
Our company has increased its sales volumes
Our company has increased its return on investment
Our company has increased its market share

## Author Contributions

**Conceptualization:** Ricardo Santa, Mario Ferrer.

**Formal analysis:** Ricardo Santa, Thomas Tegethoff.

**Investigation:** Ricardo Santa, Mario Ferrer, Thomas Tegethoff.

**Methodology:** Ricardo Santa, Mario Ferrer.

**Project administration:** Ricardo Santa.

**Resources:** Ricardo Santa.

**Software:** Ricardo Santa.

**Supervision:** Annibal Scavarda.

**Validation:** Ricardo Santa.

**Writing – original draft:** Ricardo Santa, Mario Ferrer.

**Writing – review & editing:** Ricardo Santa, Thomas Tegethoff.

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
