## [Decision Letter · Decision Letter 0]

1 Jun 2022

PONE-D-22-12655An investigation of the impact of human capital and supply chain competitive drivers on firm performance in a developing countryPLOS ONE

Dear Dr. Santa,

Thank you for submitting your manuscript to PLOS ONE. After careful consideration, we feel that it has merit but does not fully meet PLOS ONE’s publication criteria as it currently stands. Therefore, we invite you to submit a revised version of the manuscript that addresses the points raised during the review process.

We look forward to receiving your revised manuscript.

Kind regards,

Anandakumar Haldorai, PhD

Academic Editor

PLOS ONE

Journal Requirements:

4. We note you have included a table to which you do not refer in the text of your manuscript. Please ensure that you refer to Table 2 in your text; if accepted, production will need this reference to link the reader to the Table.

Additional Editor Comments:

The following issues were found:

- Sections needs to be rearranged with proper Abstract, Introduction, Literature, Proposed approach, Result and discussion, Conclusion and Reference.

- Highlights of the Study specified in abstract, it should be comes under introduction section.

- Make sure the Abstract briefly describes the paper as it is used in abstracting and citation services. Keep the Abstract between 200 words (Single paragraph). Do not use any references in the Abstract.

- Make sure that the Conclusion briefly summarizes the results of the paper it should not repeat phrases from the Introduction. Keep the Conclusion to about 200 words. Do not use any references or acronyms in the Conclusion.

Reviewers' comments:

Reviewer's Responses to Questions

**Comments to the Author**

1. Is the manuscript technically sound, and do the data support the conclusions?

Reviewer #1: Partly

Reviewer #2: Yes

2. Has the statistical analysis been performed appropriately and rigorously? 

Reviewer #1: I Don't Know

Reviewer #2: Yes

3. Have the authors made all data underlying the findings in their manuscript fully available?

Reviewer #1: No

Reviewer #2: Yes

4. Is the manuscript presented in an intelligible fashion and written in standard English?

Reviewer #1: Yes

Reviewer #2: Yes

5. Review Comments to the Author

Reviewer #1: Overall the introduction parts that build up the hypothesis is well described, while the experimental methods, results discussion and conclusion parts are weak.

Page2: 2nd paragraph: "98% of country's business"  What is the metric?

Page14-15: There are two "2.5" in section numbers

Page18: What is the industry sector that this work is focused on? Or is it completely random choice within the region? Is it possible to share the pareto of the rough industry sectors of survey responders?

Page18: What is the definition/criterion of SME and LO in this study?

Page18: How is your survey to SME/LO structured? Can you show the example set of questions?

Page18: How do the survey responders answer to the questions? Is it based on actual business performance metrics (quantitative)? Or are they based on their impression? This aspect is particularly important to how to interpret the results on page21.

Page18: How did you preprocess the survey results to feed them into SPSS/AMOS? Can you show the examples of data processing pipeline or flowchart?

Page18: Is it possible to describe a little more details on SPSS/AMOS parameter settings?

Page19: Table 3 and it's description seem to fit better after "4. Results and Discussion"

Page24: In SME if the outsourcing decision is made by personal interest of the management, where is the assumption coming from? Do you have any data or examples that can be shown? Also does "management's interest" exactly mean? If this is a major factor outsourcing-related responds in the survey seem to be invalid in this study.

Page24-25: The conclusion section is too long, and it is hard to capture the essence of the major discovery.

Reviewer #2: Thank you for the opportunity offered me to review the manuscript on “An investigation of the impact of human capital and supply chain competitive drivers on firm performance in a developing country” This summary is the observations made as I reviewed the paper.

Review Comments

Theoretically, the paper was introduced with conceptual framework theorizing the human capital and supply chain drivers in relation with firm’s performance, evidence from specific references from key literatures to support their claims.

However reference must be made of recent literatures on the topic which can easily be found and available on google scholars and other search engines.

Methodology deployed is online survey used to capture 600 firms as respondents. The SEM was used in the study was was applicable and workable enough. SPSS and AMOS software were used to analyse the data and for the various tests were performed which yielded feasible results.

The English Language though was fairly good need to be improved. Figure 2 was referenced in the results and discussions write up but it cannot be found in the paper. I suggest Figure 3 should rather be labelled as figure 2.

Fitness; The topic is good enough and fits into the scope of PLOSONE

Therefore, the manuscript need revisions on the issues raised to improve its current form for suitability for publication.

References

Besides adding additional modern references, the current references must be formatted in line with PLOSONE requirements.

6. PLOS authors have the option to publish the peer review history of their article (what does this mean?). If published, this will include your full peer review and any attached files.

Reviewer #1: No

Reviewer #2: No

---

## [Author Response · Author response to Decision Letter 0]

26 Jul 2022

Issue addressed by reviewers Answer

Page2: 2nd paragraph: "98% of country's business"  What is the metric? This was a fragment left from an earlier edit. We have removed it. 

Page14-15: There are two "2.5" in section numbers

 The section number has been adjusted to 2.6

Page18: What is the industry sector that this work is focused on? Or is it completely random choice within the region? Is it possible to share the pareto of the rough industry sectors of survey responders?

 A section with the industrial sector and professional role of the respondents was added. The section included a table that shows detailed information of the sample’s demographics

Page18: What is the definition/criterion of SME and LO in this study?

 The definition was based on the one given by the Colombian ministry of commerce. The paragraph has been adjusted accordingly. The following sentences was added : “The Colombian ministry of commerce defines the size of an organization based on the tax value unit (TVU), which is a value equivalent to the Colombian peso used to determine different tax obligations, such as the minimum amounts of withholding at source or penalties. Organizations with a TVU below 1.736.565 are considered SME. All other organizations are ranked as LO.”

Page18: How is your survey to SME/LO structured? Can you show the example set of questions?

 The Measurement items are shown in Appendix A.

Page18: How do the survey responders answer to the questions? Is it based on actual business performance metrics (quantitative)? Or are they based on their impression? This aspect is particularly important to how to interpret the results on page21.

 The required information was added:

“Statements related to the operationalization of the various constructs were evaluated based on a five-point Likert-type scale (Strongly Disagree—Strongly Agree) representing the perception of the respondents according to the different questions”

Page18: How did you preprocess the survey results to feed them into SPSS/AMOS? Can you show the examples of data processing pipeline or flowchart?

 One of the advantages of online survey tools is the ability to transfer the data collected directly to SPSS. We have clarified this thus: 

Given the advantages of online surveys (Evans, 2005), we invited potential respondents to participate by email, with a link to access the questionnaire in Microsoft Forms, from which the data can be transferred directly to SPSS.

Page18: Is it possible to describe a little more details on SPSS/AMOS parameter settings?

 The following section in pages 13 and 14 were improved to address the reviewer’s suggestion. 

We used SPSS (IBM Corp) and AMOS software to analyze the data. The analysis stage included confirmation of the conceptualized model shown in Figure 1. The following steps summarize the statistical analysis performed to estimate the model predictiveness as well as its indices.

1. Confirmatory factor analysis (CFA) with the following settings: 

• Discrepancy: Asymptotically distribution-free

Covariances supplied as input 

• Unbiased

Covariances to be analyzed 

• Maximum likelihood

2. Factor loading estimation (assessment of the relationship between the observed and continuous latent variables)

3. Internal consistency (Cronbach’s alpha coefficient shown in table 2) (Cooksey, 2007; Hair et al., 2010).

4. Testing of construct validity (CFA)

Page19: Table 3 and it's description seem to fit better after "4. Results and Discussion"

 The section containing Tables 2 and 3 was transferred to section 4. Results and discussion

Page24: In SME if the outsourcing decision is made by personal interest of the management, where is the assumption coming from? Do you have any data or examples that can be shown? Also does "management's interest" exactly mean? If this is a major factor outsourcing-related responds in the survey seem to be invalid in this study.

 Nowhere was this given as a major outsourcing factor. It is mentioned in the conclusion as a possible reason for the results of the study. To clarify our meaning, we have edited the sentence thus:

It is possible that in Colombian SMEs, outsourcing decisions are made based on the personal interests of the management team.

Page24-25: The conclusion section is too long, and it is hard to capture the essence of the major discovery.

 We have not edited this section as we believe that removing any of the content would detract from the conclusions drawn.

Theoretically, the paper was introduced with conceptual framework theorizing the human capital and supply chain drivers in relation with firm’s performance, evidence from specific references from key literatures to support their claims.

However reference must be made of recent literatures on the topic which can easily be found and available on google scholars and other search engines.

 Following references were added to the document:

- Matthyssens, P. (2019), “Reconceptualizing value innovation for Industry 4.0 and the Industrial Internet of Things”, Journal of Business & Industrial Marketing, Vol. 34 No. 6, pp. 1203-1209. https://doi.org/10.1108/JBIM-11-2018-0348”

- Falahat, M., Ramayah, T., Soto-Acosta, P., & Lee, Y.-Y. (2020). SMEs internationalization: The role of product innovation, market intelligence, pricing and marketing communication capabilities as drivers of SMEs’ international performance. Technological Forecasting and Social Change, 152, 119908. https://doi.org/https://doi.org/10.1016/j.techfore.2020.119908”

- Hussain S, Xuetong W, Hussain T. Impact of Skilled and Unskilled Labor on Project Performance Using Structural Equation Modeling Approach. SAGE Open. January 2020. doi:10.1177/2158244020914590

- Manzoor, F., Wei, L., Bányai, T., Nurunnabi, M., & Subhan, Q. A. (2019). An Examination of Sustainable HRM Practices on Job Performance: An Application of Training as a Moderator. Sustainability, 11(8), 2263.

- Ferreira, J., Coelho, A., & Moutinho, L. (2020). Dynamic capabilities, creativity and innovation capability and their impact on competitive advantage and firm performance: The moderating role of entrepreneurial orientation. Technovation, 92-93, 102061. https://doi.org/https://doi.org/10.1016/j.technovation.2018.11.004

- Ho, H, Kuvaas, B. Human resource management systems, employee well-being, and firm performance from the mutual gains and critical perspectives: The well-being paradox. Hum Resour Manage. 2020; 59: 235– 253. https://doi.org/10.1002/hrm.21990

- Alzabari, S. A. H., Talab, H. R., & Flayyih, H. H. (2019). The effect of internal training and auditing of auditors on supply chain management: An empirical study in listed companies of Iraqi stock exchange for the period 2012-2015. Int. J Sup. Chain. Mgt Vol, 8(5), 1070.

- Koot, M., Mes, M. R. K., & Iacob, M. E. (2021). A systematic literature review of supply chain decision making supported by the Internet of Things and Big Data Analytics. Computers & Industrial Engineering, 154, 107076. https://doi.org/https://doi.org/10.1016/j.cie.2020.107076

- Clauss, T., Abebe, M., Tangpong, C., & Hock, M. (2019). Strategic agility, business model innovation, and firm performance: an empirical investigation. IEEE transactions on engineering management, 68(3), 767-784.

- Joiner B. Leadership Agility for Organizational Agility. Journal of Creating Value. 2019;5(2):139-149. doi:10.1177/2394964319868321

- Al Humdan, E., Shi, Y., Behnia, M. and Najmaei, A. (2020), "Supply chain agility: a systematic review of definitions, enablers and performance implications", International Journal of Physical Distribution & Logistics Management, Vol. 50 No. 2, pp. 287-312. https://doi.org/10.1108/IJPDLM-06-2019-0192

- Abdallah, A.B., Alfar, N.A. and Alhyari, S. (2021), "The effect of supply chain quality management on supply chain performance: the indirect roles of supply chain agility and innovation", International Journal of Physical Distribution & Logistics Management, Vol. 51 No. 7, pp. 785-812. https://doi.org/10.1108/IJPDLM-01-2020-0011

- Dev, N. K., Shankar, R., & Qaiser, F. H. (2020). Industry 4.0 and circular economy: Operational excellence for sustainable reverse supply chain performance. Resources, Conservation and Recycling, 153, 104583.

- Baah, C., Agyeman, D. O., Acquah, I. S. K., Agyabeng-Mensah, Y., Afum, E., Issau, K., ... & Faibil, D. (2021). Effect of information sharing in supply chains: understanding the roles of supply chain visibility, agility, collaboration on supply chain performance. Benchmarking: An International Journal, 29(2), 434-455.

- Reschke, Jan, and Sergio Gallego-García. 2021. "A Novel Methodology for Assessing and Modeling Manufacturing Processes" Applied Sciences 11, no. 21: 10117. https://doi.org/10.3390/app112110117

- Wydyanto, W., & Ilhamalimy, R. R. (2021). Determination of Purchasing Decisions and Customer Satisfaction: Analysis of Service Quality and Product Quality (Marketing Management Literature Review). Dinasti International Journal of Education Management And Social Science, 2(3), 565-575.

- Victor DeMiguel, Alberto Martín-Utrera, Francisco J Nogales, Raman Uppal, A Transaction-Cost Perspective on the Multitude of Firm Characteristics, The Review of Financial Studies, Volume 33, Issue 5, May 2020, Pages 2180–2222, https://doi.org/10.1093/rfs/hhz085

- Tsay, A. A., Gray, J. V., Noh, I. J., & Mahoney, J. T. (2018). A review of production and operations management research on outsourcing in supply chains: Implications for the theory of the firm. Production and Operations Management, 27(7), 1177-1220.

The English Language though was fairly good need to be improved.

 The manuscript was reviewed by a native English speaker who works for Proofed Inc. 

Figure 2 was referenced in the results and discussions write up but it cannot be found in the paper. I suggest Figure 3 should rather be labelled as figure 2.

 The figure numbers were corrected accordingly. The final structural equation models were labelled as Fig 2 and Fig 3 for SME and LO, respectively

---

## [Decision Letter · Decision Letter 1]

15 Aug 2022

PONE-D-22-12655R1An investigation of the impact of human capital and supply chain competitive drivers on firm performance in a developing countryPLOS ONE

Dear Dr. Santa,

Thank you for submitting your manuscript to PLOS ONE. After careful consideration, we feel that it has merit but does not fully meet PLOS ONE’s publication criteria as it currently stands. Therefore, we invite you to submit a revised version of the manuscript that addresses the points raised during the review process. Please submit your revised manuscript by Sep 29 2022 11:59PM. If you will need more time than this to complete your revisions, please reply to this message or contact the journal office at plosone@plos.org. Please include the following items when submitting your revised manuscript:A rebuttal letter that responds to each point raised by the academic editor and reviewer(s). You should upload this letter as a separate file labeled 'Response to Reviewers'.A marked-up copy of your manuscript that highlights changes made to the original version. You should upload this as a separate file labeled 'Revised Manuscript with Track Changes'.An unmarked version of your revised paper without tracked changes. You should upload this as a separate file labeled 'Manuscript'.If applicable, we recommend that you deposit your laboratory protocols in protocols.io to enhance the reproducibility of your results. Protocols.io assigns your protocol its own identifier (DOI) so that it can be cited independently in the future. For instructions see: https://journals.plos.org/plosone/s/submission-guidelines#loc-laboratory-protocols. Additionally, PLOS ONE offers an option for publishing peer-reviewed Lab Protocol articles, which describe protocols hosted on protocols.io. Read more information on sharing protocols at https://plos.org/protocols?utm_medium=editorial-email&utm_source=authorletters&utm_campaign=protocols.

We look forward to receiving your revised manuscript.

Kind regards,

Anandakumar Haldorai, PhD

Academic Editor

PLOS ONE

Journal Requirements:

Additional Editor Comments (if provided):

Please carefully address the issues raised in the comments and, up front in your revised paper.

Make sure the Abstract briefly describes the paper as it is used in abstracting and citation services. Clearly specify the Purpose, Methodology, and problem findings.

Verify the already published papers in he journal website and clearly prepare the contents according to the standard.

Ex. abstract section not properly developed.

Subsections are not properly numbered,

Paper not properly aligned, Image and table qualities are very poor.

Include a list of six to ten key words after the Abstract.

Spell out each acronym the first time used in the body of the paper. Spell out acronyms in the Abstract only if used there.

You may ignore any suggestion of including self-references by reviewers if not applicable. Follow the proper journal reference format.

Include a paragraph at the end of the Introduction describing the organization of the paper.

Make sure that the Conclusion briefly summarizes the results of the paper it should not repeat phrases from the Introduction. Keep the Conclusion to about 200 words. Do not use any references or acronyms in the Conclusion.

Make sure all figures and tables are referred to in the body of the paper.

It is recommended to use a professional proofread and native English correction. Papers with less than excellent English will not be published even if technically perfect. Complete native correction is recommended

Reviewers' comments:

Reviewer's Responses to Questions

**Comments to the Author**

1. If the authors have adequately addressed your comments raised in a previous round of review and you feel that this manuscript is now acceptable for publication, you may indicate that here to bypass the “Comments to the Author” section, enter your conflict of interest statement in the “Confidential to Editor” section, and submit your "Accept" recommendation.

Reviewer #1: All comments have been addressed

Reviewer #2: All comments have been addressed

2. Is the manuscript technically sound, and do the data support the conclusions?

Reviewer #1: Yes

Reviewer #2: Yes

3. Has the statistical analysis been performed appropriately and rigorously? 

Reviewer #1: Yes

Reviewer #2: Yes

4. Have the authors made all data underlying the findings in their manuscript fully available?

Reviewer #1: Yes

Reviewer #2: Yes

5. Is the manuscript presented in an intelligible fashion and written in standard English?

Reviewer #1: Yes

Reviewer #2: Yes

6. Review Comments to the Author

Reviewer #1: The detailed premise and background of the survey and statistical analysis are extensively reinforced, and thus I now have more confidence that the literature is more proper to be published. Congratulations on the successful modifications.

Reviewer #2: Review Comments

Manuscript Number: PONE-D-22-12655R1

The manuscript on “An investigation of the impact of human capital and supply chain competitive drivers on firm performance in a developing country” is currently revised to improve the previous issues raised and the comments were adequately catered for in the modification.

The manuscript is modified based on the issues raised in the previous review comments and corrected them as follows;

Adequate recent literature on the topic are now added in the reviewed version;

Additional references are added and formatted in line with PLOSONE requirements;

The English Language is good and acceptable;

The previous wrong numbering of the figures are now corrected to be okay;

The manuscript is now revised and improved in its current form to be suitable for publication.

The topic fits into the scope of PLOSONE. The paper employed a conceptual framework theorizing the human capital and supply chain drivers in relation with firm’s performance, evidence from specific references from key literatures were used to support the claims.

The Methodology deployed is online survey used to capture 600 firms as respondents. The SEM was used in the study was was applicable and workable enough. SPSS and AMOS software were used to analyse the data and for the various tests were performed which yielded feasible results.

Therefore, I recommend it for publication in PLOSONE journal

7. PLOS authors have the option to publish the peer review history of their article (what does this mean?). If published, this will include your full peer review and any attached files.

Reviewer #1: No

Reviewer #2: **Yes: **STEPHEN OKYERE (PhD, MBA. PGD, B.ED, CMILT, MCISCM)

---

## [Author Response · Author response to Decision Letter 1]

28 Aug 2022

PONE-D-22-12655R1

An investigation of the impact of human capital and supply chain competitive drivers on firm performance in a developing country

PLOS ONE

Issue addressed by reviewers /Answer

Please review your reference list to ensure that it is complete and correct. If you have cited papers that have been retracted, please include the rationale for doing so in the manuscript text, or remove these references and replace them with relevant current references. Any changes to the reference list should be mentioned in the rebuttal letter that accompanies your revised manuscript. If you need to cite a retracted article, indicate the article’s retracted status in the References list and also include a citation and full reference for the retraction notice. All references were reviewed to ensure that they are complete and correct.

No retracted papers were cited. 

No changes to the reference list are needed.

Make sure the Abstract briefly describes the paper as it is used in abstracting and citation services. Clearly specify the Purpose, Methodology, and problem findings. The abstract already has the following sections:

Purpose, Design, Findings, and Practical implications 

Verify the already published papers in the journal website and clearly prepare the contents according to the standard.

Ex. abstract section not properly developed.

Subsections are not properly numbered,

Paper not properly aligned, Image and table qualities are very poor.

 We checked several published articles.

Include a list of six to ten keywords after the Abstract. The article has 8 keywords. 

Spell out each acronym the first time used in the body of the paper. Spell out acronyms in the Abstract only if used there. All acronyms are correctly spelled out. 

You may ignore any suggestion of including self-references by reviewers if not applicable. Follow the proper journal reference format. We followed all suggestions addressed by the reviewers.

We followed the journal`s reference format.

Include a paragraph at the end of the Introduction describing the organization of the paper. The article already has a paragraph describing the organization of the paper: 

The rest of the paper is organized as follows. Section 2 provides a conceptual framework used for the formulation of hypotheses and the presentation of the conceptual model of the relationship between the proposed variables, while section 3 presents the research method. The fourth section presents the most important results and findings. Section 5 shows the measurement items used. Finally, in section 6, the conclusions of the study, its limitations, and future lines of research are discussed.

Make sure that the Conclusion briefly summarizes the results of the paper it should not repeat phrases from the Introduction. Keep the Conclusion to about 200 words. Do not use any references or acronyms in the Conclusion. We removed a section in the conclusion that did not add to the discussion. 

We strongly believe that the rest of the conclusion should be retained as the model explored in the article has 12 hypotheses, all of which have relevant lessons. 

Make sure all figures and tables are referred to in the body of the paper. All figures and tables were checked. 

It is recommended to use a professional proofread and native English correction. Papers with less than excellent English will not be published even if technically perfect. Complete native correction is recommended The manuscript was reviewed by a native English speaker who works for Proofed Inc.

---

## [Editor Report · Decision Letter 2]

1 Sep 2022

An investigation of the impact of human capital and supply chain competitive drivers on firm performance in a developing country

PONE-D-22-12655R2

Dear Dr. Santa,

We’re pleased to inform you that your manuscript has been judged scientifically suitable for publication and will be formally accepted for publication once it meets all outstanding technical requirements.

Kind regards,

Anandakumar Haldorai, PhD

Academic Editor

PLOS ONE

Additional Editor Comments (optional):

Recommended for further process.
---

## [Editor Report · Acceptance letter]

9 Nov 2022

PONE-D-22-12655R2 

An investigation of the impact of human capital and supply chain competitive drivers on firm performance in a developing country 

Dear Dr. Santa:

I'm pleased to inform you that your manuscript has been deemed suitable for publication in PLOS ONE. Congratulations! Your manuscript is now with our production department. 

Kind regards, 

on behalf of

Dr. Anandakumar Haldorai 

Academic Editor

PLOS ONE